# Conditional Diffusion Process for Inverse Halftoning

**Hao Jiang**
Peking University
jianghao@stu.pku.edu.cn

**Yadong Mu**[*]
Peking University
Peng Cheng Laboratory
myd@pku.edu.cn

## Abstract

Inverse halftoning is a technique used to recover realistic images from ancient prints (*e.g.*, photographs, newspapers, books). The rise of deep learning has led to the gradual incorporation of neural network designs into inverse halftoning methods. Most of existing inverse halftoning approaches adopt the U-net architecture, which uses an encoder to encode halftone prints, followed by a decoder for image reconstruction. However, the mainstream supervised learning paradigm with element-wise regression commonly adopted in U-net based methods has poor generalization ability in practical applications. Specifically, when there is a large gap between the dithering patterns of the training and testing halftones, the reconstructed continuous-tone images have obvious artifacts. This is an important issue in practical applications, since the algorithms for generating halftones are ever-evolving. Even for the same algorithm, different parameter choices will result in different halftone dithering patterns. In this paper, we propose the first generative halftoning method in the literature, which regards the black pixels in halftones as physically moving particles, and makes the randomly distributed particles move under some certain guidance through reverse diffusion process, so as to obtain desired halftone patterns. In particular, we propose a Conditional Diffusion model for image Halftoning (CDH), which consists of a halftone dithering process and an inverse halftoning process. By changing the initial state of the diffusion model, our method can generate visually plausible halftones with different dithering patterns under the condition of image gray level and Laplacian prior. To avoid introducing redundant patterns and undesired artifacts, we propose a meta-halftone guided network to incorporate blue noise guidance in the diffusion process. In this way, halftone images subject to more diverse distributions are fed into the inverse halftoning model, which helps the model to learn a more robust mapping from halftone distributions to continuous-tone distributions, thereby improving the generalization ability to unseen samples. Quantitative and qualitative experimental results demonstrate that the proposed method achieves state-of-the-art results.

## 1 Introduction

Halftoning refers to the task of simulating the brightness change of a continuous-tone image by changing the size or frequency of halftone dots (such as ink dots). In the last century, halftoning technology has been widely used in the printing industry to store precious image data through old newspapers, books, photographs, *etc*. At the same time, inverse halftoning technology emerged to recover stored continuous-tone image from vintage materials. The goal of the inverse halftoning technique is to minimize the loss of information in the restoration process, so that the restored print has the highest possible visual quality.

---

[*]Corresponding author.

36th Conference on Neural Information Processing Systems (NeurIPS 2022).

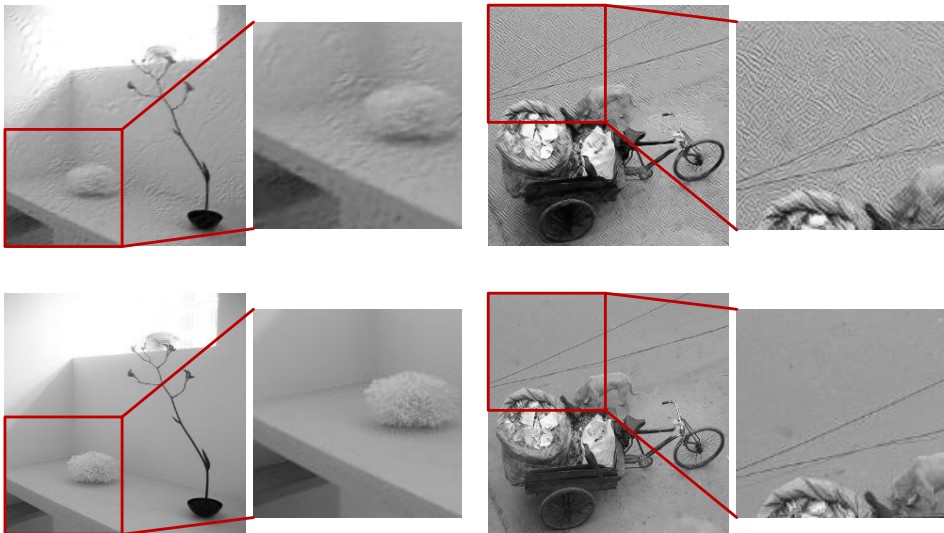

Figure 1: Illustration of the presence of artifacts in restored images by previous inverse halftoning method (Xia and Wong, 2018), where the gap exists between the dithering patterns of training and testing halftones. The top row shows inverse halftoning results for some testing samples, and the bottom row demonstrates corresponding ground-truth continuous-tone images.

Traditional inverse halftoning methods are mainly based on image filtering (*e.g.*, wavelet domain filtering (Xiong et al., 1999), edge-preserving filtering (Kite et al., 2000), SUSAN filtering (Siddiqui and Bouman, 2007)), bilateral filtering (Sun et al., 2014) and statistical learning methods (*e.g.*, least mean square filtering (Chen and Hang, 1997), maximum a posteriori (Stevenson, 1997), look-up table (Mese and Vaidyanathan, 2001; Chung and Wu, 2005), dictionary learning (Zhang et al., 2018)). With the revival of deep learning technology, inverse halftoning methods based on deep neural networks have made great progress and attracted more and more attention (Hou and Qiu, 2017; Xiao et al., 2017; Kim and Park, 2018). The most representative method is the U-net (Ronneberger et al., 2015) based architecture, which has an encoder to learn the hidden encoding of the halftone printing, followed by a decoder to reconstruct the image (Xia and Wong, 2018; Gao et al., 2019).

However, the paradigm of supervised learning with element-wise regression commonly adopted in the U-net based methods suffers from poor generalization in practical applications. Specifically, when there is a large gap between the dithering patterns of training and testing halftones, the restored continuous-tone images often have obvious artifacts. This is an important issue in applications, since the algorithms used for generating halftones evolve with time. Even for the same algorithm, different parameter choices will result in different halftone dithering patterns. Taking frequency modulation (FM) halftoning as an example, we select 9 classical error diffusion processes with different dithering patterns, namely Floyd-Steinberg Dithering, Jarvis-Judice-Ninke Dithering, Stucki Dithering, Atkinson Dithering, Burkes Dithering, Sierra Dithering, and several of their variants (Lau and Arce, 2018). We train the U-net model on the halftones generated by 5 of the algorithms, and test on the halftones obtained with the rest 4 algorithms. The experimental results are shown in Figure 1. Artifacts can be clearly observed in the restored continuous-tone images.

To address this problem, in this paper, we propose a Conditional Diffusion model for image Halftoning (CDH), as shown in Figure 2, which consists of a halftone dithering process and an inverse halftoning process. We regard the black pixels in halftones as physically moving particles, and make the randomly distributed particles move under some certain guidance through the reverse diffusion process, so as to obtain the desired halftone distribution. Specifically, for the halftone dithering process, we train a conditional diffusion model to generate halftones with different dithering patterns under the condition of image gray level and Laplacian prior. By changing the initial state of the diffusion model, it can simulate different dithering processes to generate diverse halftone images. To avoid introducing redundant patterns and undesired artifacts during halftone generation, we propose a meta-halftone guided network to incorporate the blue noise guidance into the diffusion process. For inverse halftoning, we train an inverse halftoning diffusion model to learn the mapping function from

the halftone distribution to the continuous-tone distribution. In this way, halftones subject to more diverse distributions are input to the inverse halftoning model, which helps the model to learn a more robust mapping and improve the generalization ability to unseen samples.

Our contributions are summarized as follows:

- This is the first work to propose a generative halftoning method, which regards the black pixels in halftones as physically moving particles, and makes the randomly distributed particles move under some certain guidance through the reverse diffusion process, so as to obtain the desired halftone dithering patterns.
- To avoid introducing redundant patterns and undesired artifacts during halftone generation, we propose a meta-halftone guided network to incorporate the blue noise guidance into the halftone diffusion process.
- To obtain better generalization ability, we use the $x_0$ state of halftone dithering diffusion as the condition for inverse halftoning diffusion, so that the inverse halftoning diffusion model benefits from a wider range of dithering patterns and learns a more robust mapping.

We conduct experiments on the dataset consisting of 9 halftoning algorithms, and quantitative and qualitative experiments demonstrate that the proposed method achieves state-of-the-art results.

## 2   Related Work

**Inverse Halftoning.** Traditional inverse halftoning methods are mainly based on image filtering (*e.g.*, edge-preserving filtering (Kite et al., 2000), wavelet domain filtering (Xiong et al., 1999), bilateral filtering (Sun et al., 2014), SUSAN filtering (Siddiqui and Bouman, 2007)) and statistical learning methods (*e.g.*, look-up table (Chung and Wu, 2005; Mese and Vaidyanathan, 2001), least mean square filtering (Chen and Hang, 1997), maximum a posteriori (Stevenson, 1997), dictionary learning (Zhang et al., 2018)). For example, Sun *et al.* (Sun et al., 2014) proposed to use an anisotropic Gaussian filter and an edge-preserving filter for inverse halftoning. With the revival of deep learning technology, some researchers have tried to use deep neural networks to accomplish inverse halftoning, *e.g.*, U-net based models (Hou and Qiu, 2017; Xiao et al., 2017; Gao et al., 2019), residual learning based models (Xia and Wong, 2018) and contextual learning based models (Kim and Park, 2018). For example, Xia *et al.* (Xia and Wong, 2018) proposed a progressively residual based U-net that synthesizes the global tone and subtle details to generate inverse halftones. However, these approaches suffer from unacceptable artifacts in the restored continuous-tone images when faced with dithering pattern gaps.

**Diffusion Models.** Diffusion models are a class of deep generative models developed from non-equilibrium thermodynamics (Sohl-Dickstein et al., 2015). They define a Markovian process for the diffusion steps, incrementally adding random noises to the original data, and then learn the reverse diffusion process to resample the data from noises (Sohl-Dickstein et al., 2015; Ho et al., 2020; Luo and Hu, 2021). Diffusion models are closely related to score-based generative models, which generate samples by Langevin dynamics based on estimated gradients of the data distribution (Song and Ermon, 2019, 2020; Song et al., 2020b, 2021). Some improved techniques are proposed to help the diffusion model converge to a lower negative log-likelihood or to speed up sampling (Nichol and Dhariwal, 2021; Dhariwal and Nichol, 2021; Song et al., 2020a). For example, Song *et al.* (Song et al., 2020a) generalizes denoising diffusion probabilistic models via a class of non-Markovian processes, which can correspond to deterministic generative processes and gives rise to implicit models that produce high quality samples much faster. However, these diffusion models cannot be directly applied to halftone generation, since they do not take into account the blue noise properties of halftone prints, making it difficult to generate visually pleasing dithering patterns.

## 3   Method

### 3.1   Halftone Dithering Diffusion Conditioned on Gray-scale and Laplacian Prior

Traditional halftone dithering methods are mainly based on techniques of amplitude modulation (Blatner and Roth, 1993; Campbell et al., 1966) and frequency modulation (Eschbach, 1997; Floyd, 1976; Meşe and Vaidyanathan, 1999). With the development of deep learning, some modern approaches try

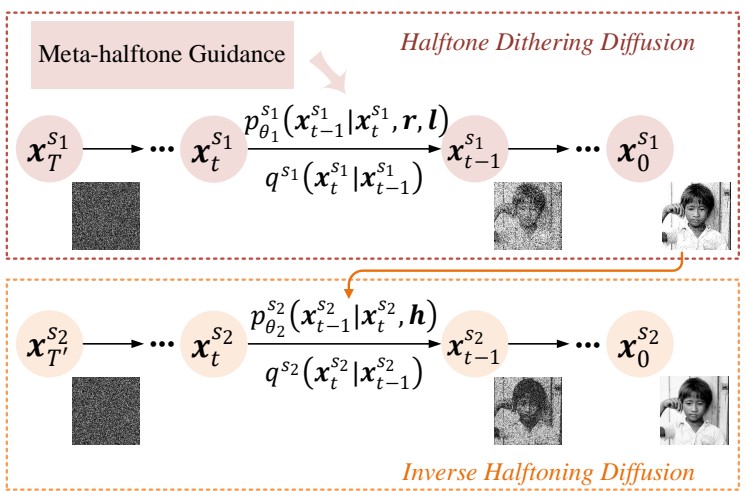

Figure 2: Illustration of the proposed Conditional Diffusion model for image Halftoning (CDH), which consists of a halftone dithering diffusion process and an inverse halftoning diffusion process.

to generate halftone images using convolutional neural networks (Kim and Park, 2018; Xia et al., 2021). However, these methods are limited to specific dithering patterns and cannot achieve flexible halftone generation. To address this problem, considering that the halftone prints are only composed of black and white pixels, we regard the black pixels in halftones as moving particles, and make these particles move from random Gaussian distributions to halftone distributions under reverse diffusion processes, so as to achieve a generative halftoning approach.

Given the distribution $q^{s_1}(\boldsymbol{x}_0^{s_1})$ of halftone prints, the diffusion process $q^{s_1}$ is a Markovian noising process (Ho et al., 2020) that gradually adds noise to $\boldsymbol{x}_0^{s_1}$ to obtain $\boldsymbol{x}_{1:T}^{s_1}$, where $s_1$ denotes the halftone dithering diffusion stage. Specifically, at each step $t$, the diffusion step adds the random Gaussian noise with a $\beta_t$-controlled variance:

$$q^{s_1}(\boldsymbol{x}_{1:T}^{s_1}|\boldsymbol{x}_0^{s_1}) = \prod_{t=1}^{T} q^{s_1}(\boldsymbol{x}_t^{s_1}|\boldsymbol{x}_{t-1}^{s_1}), \tag{1}$$

$$q^{s_1}(\boldsymbol{x}_t^{s_1}|\boldsymbol{x}_{t-1}^{s_1}) = \mathcal{N}(\boldsymbol{x}_t^{s_1}; \sqrt{1-\beta_t}\boldsymbol{x}_{t-1}^{s_1}, \beta_t\mathbf{I}), \tag{2}$$

where $\beta_t \in (0,1)$, $t = 1, ..., T$. With the reparameterization trick (Kingma and Welling, 2013), we can sample $\boldsymbol{x}_t^{s_1}$ from any time step $t$ in a closed form: $\boldsymbol{x}_t^{s_1} = \sqrt{\bar{\alpha}_t}\boldsymbol{x}_0^{s_1} + \sqrt{1-\bar{\alpha}_t}\boldsymbol{\epsilon}$, $\boldsymbol{\epsilon} \sim \mathcal{N}(\mathbf{0}, \mathbf{I})$, that is

$$q^{s_1}(\boldsymbol{x}_t^{s_1}|\boldsymbol{x}_0^{s_1}) = \mathcal{N}(\boldsymbol{x}_t^{s_1}; \sqrt{\bar{\alpha}_t}\boldsymbol{x}_0^{s_1}, (1-\bar{\alpha}_t)\mathbf{I}), \tag{3}$$

where $\alpha_t = 1 - \beta_t$, $\bar{\alpha}_t = \prod_{i=1}^{t} \alpha_i$. In this way, we can directly derive $\boldsymbol{x}_t^{s_1}$ by $q^{s_1}(\boldsymbol{x}_t^{s_1}|\boldsymbol{x}_0^{s_1})$ without repeatedly applying the Markovian process $q^{s_1}$ and calculating $q^{s_1}(\boldsymbol{x}_t^{s_1}|\boldsymbol{x}_{t-1}^{s_1})$.

In the halftone dithering scenario, we would like to obtain the halftone sample $\boldsymbol{x}_0^{s_1}$ via the reverse diffusion process $q^{s_1}(\boldsymbol{x}_{t-1}^{s_1}|\boldsymbol{x}_t^{s_1})$ from a random Gaussian distribution, *i.e.*, the state $\boldsymbol{x}_T^{s_1} \sim \mathcal{N}(\mathbf{0}, \mathbf{I})$. Considering that the particles tend to rely on the gradient information of images when forming halftone patterns, we use the image gray level $\boldsymbol{r}$ and the Laplacian prior $\boldsymbol{l}$ as conditions to guide the reverse diffusion process. Formally, the reverse halftoning diffusion process is defined on the distribution $p_{\theta_1}^{s_1}(\boldsymbol{x}_{0:T}^{s_1}|\boldsymbol{r}, \boldsymbol{l})$, which is a Markov chain with start state $p^{s_1}(\boldsymbol{x}_T^{s_1}) = \mathcal{N}(\boldsymbol{x}_T^{s_1}; \mathbf{0}, \mathbf{I})$:

$$p_{\theta_1}^{s_1}(\boldsymbol{x}_{0:T}^{s_1}|\boldsymbol{r}, \boldsymbol{l}) = p^{s_1}(\boldsymbol{x}_T^{s_1}) \prod_{t=1}^{T} p_{\theta_1}^{s_1}(\boldsymbol{x}_{t-1}^{s_1}|\boldsymbol{x}_t^{s_1}, \boldsymbol{r}, \boldsymbol{l}), \tag{4}$$

$$p_{\theta_1}^{s_1}(\boldsymbol{x}_{t-1}^{s_1}|\boldsymbol{x}_t^{s_1}, \boldsymbol{r}, \boldsymbol{l}) = \mathcal{N}(\boldsymbol{x}_{t-1}^{s_1}; \boldsymbol{\mu}_{\theta_1}^{s_1}(\boldsymbol{x}_t^{s_1}, \boldsymbol{r}, \boldsymbol{l}, t), \boldsymbol{\Sigma}_{\theta_1}^{s_1}(\boldsymbol{x}_t^{s_1}, \boldsymbol{r}, \boldsymbol{l}, t)). \tag{5}$$

Different from previous methods like using class labels (Dhariwal and Nichol, 2021) or shape latents (Luo and Hu, 2021) as conditions for the diffusion model, we use the pixel-wise image priors

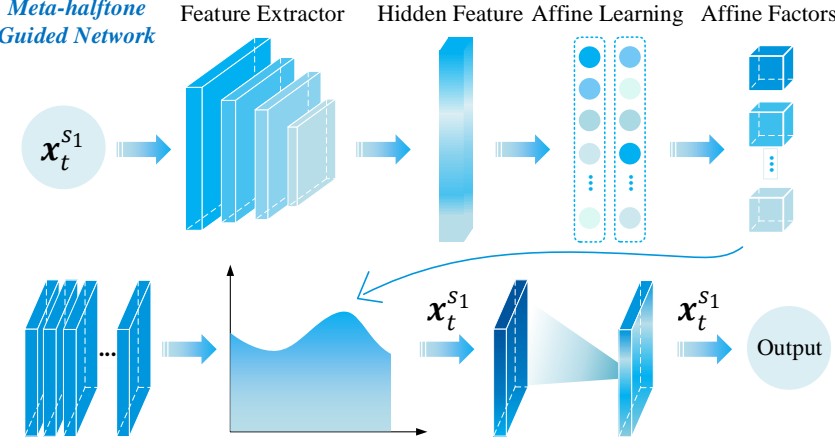

Figure 3: Illustration of the proposed Meta-halftone Guided Network.

as conditions. The conditions are performed in a simple way, *i.e.*, we concatenate the state vector $\boldsymbol{x}_t^{s_1}$ at diffusion step $t$ with $\boldsymbol{r}$ and $\boldsymbol{l}$ along the channel dimension to provide pixel-wise guidance:

$$\boldsymbol{x}_t^{s_1} := \boldsymbol{x}_t^{s_1} \oplus \boldsymbol{r} \oplus \boldsymbol{l}, \tag{6}$$

where $\oplus$ denotes the concatenate operation. Due to the properties of halftones, local dithering patterns are more important for generating high-quality halftones than global semantic information. We also observe in experiments that excessive introduction of global information can be harmful to dithering results.

We would like to learn the model $p_{\theta_1}^{s_1}$ to approximate the conditional probabilities $q^{s_1}(\boldsymbol{x}_{t-1}^{s_1}|\boldsymbol{x}_t^{s_1}, \boldsymbol{x}_0^{s_1})$. With Bayes' theorem, it has

$$q^{s_1}(\boldsymbol{x}_{t-1}^{s_1}|\boldsymbol{x}_t^{s_1}, \boldsymbol{x}_0^{s_1}) = q^{s_1}(\boldsymbol{x}_t^{s_1}|\boldsymbol{x}_{t-1}^{s_1}, \boldsymbol{x}_0^{s_1}) \frac{q^{s_1}(\boldsymbol{x}_{t-1}^{s_1}|\boldsymbol{x}_0^{s_1})}{q^{s_1}(\boldsymbol{x}_t^{s_1}|\boldsymbol{x}_0^{s_1})}, \tag{7}$$

and $q^{s_1}(\boldsymbol{x}_{t-1}^{s_1}|\boldsymbol{x}_t^{s_1}, \boldsymbol{x}_0^{s_1})$ can be represented as a Gaussian distribution:

$$q^{s_1}(\boldsymbol{x}_{t-1}^{s_1}|\boldsymbol{x}_t^{s_1}, \boldsymbol{x}_0^{s_1}) = \mathcal{N}(\boldsymbol{x}_{t-1}^{s_1}; \tilde{\boldsymbol{\mu}}_t^{s_1}(\boldsymbol{x}_t^{s_1}, \boldsymbol{x}_0^{s_1}), \tilde{\beta}_t\mathbf{I}), \tag{8}$$

$$\tilde{\boldsymbol{\mu}}_t^{s_1}(\boldsymbol{x}_t^{s_1}, \boldsymbol{x}_0^{s_1}) = \frac{\sqrt{\bar{\alpha}_{t-1}}\beta_t}{1-\bar{\alpha}_t}\boldsymbol{x}_0^{s_1} + \frac{\sqrt{\alpha_t}(1-\bar{\alpha}_{t-1})}{1-\bar{\alpha}_t}\boldsymbol{x}_t^{s_1}, \tilde{\beta}_t = \frac{1-\bar{\alpha}_{t-1}}{1-\bar{\alpha}_t}\beta_t. \tag{9}$$

The simplified objective function (Ho et al., 2020) is used to train the model during the halftone dithering diffusion process:

$$L^{s_1} = \mathbb{E}_{t\sim[1,T],\boldsymbol{x}_0^{s_1}\sim q^{s_1}(\boldsymbol{x}_0^{s_1}),\boldsymbol{\epsilon}\sim\mathcal{N}(\mathbf{0},\mathbf{I})}\big[||\boldsymbol{\epsilon} - \boldsymbol{\epsilon}_{\theta_1}^{s_1}(\sqrt{\bar{\alpha}_t}\boldsymbol{x}_0^{s_1} + \sqrt{1-\bar{\alpha}_t}\boldsymbol{\epsilon}, t)||^2\big]. \tag{10}$$

### 3.2 Meta-halftone Guided Network

Blue noise properties are critical for generating high-quality halftones. It avoids noticeable low-frequency visual artifacts in the resulting halftones by forcing random pixel dithering (Lau and Arce, 2018). Some traditional methods have been proposed to derive blue noise dithering patterns, such as simulated annealing based methods (Sullivan et al., 1991), void-and-cluster techniques (Ulichney, 1993), power spectrum manipulation algorithms (Yao and Parker, 1994), and dither pattern ordering methods (Lau et al., 1999). However, most of these approaches are based on statistical methods or manually designed blue-noise dithering matrices, which are difficult to directly apply to learning-based neural frameworks. Recently, some researchers have tried to inject blue noise properties into the L1 norm for learning objectives (Xia et al., 2021), however, since the calculation of the L1 norm requires known halftone generation results, it cannot be used in generative models. So far, how to

incorporate the blue-noise dithering properties into halftone generative models remains an unexplored problem.

To address this issue, we propose a meta-halftone guided network (as shown in Figure 3), which introduces blue noise guidance into halftone dithering diffusion process and avoids dithering results containing artifacts or redundant patterns. Formally, for step $t$ of the halftone dithering diffusion process, we consider the state vector $\boldsymbol{x}_t^{s_1}$ and take it as input to the meta-halftone guided network. We first feed $\boldsymbol{x}_t^{s_1}$ into a feature extraction network $\mathbf{E}$ to obtain the extracted hidden feature $\boldsymbol{f}(\boldsymbol{x}_t^{s_1})$, $i.e.$, $\boldsymbol{f}(\boldsymbol{x}_t^{s_1}) = \mathbf{E}(\boldsymbol{x}_t^{s_1})$. We use the pre-trained VGG network (Simonyan and Zisserman, 2015) as $\mathbf{E}$ in the experiment, other networks such as InceptionNet (Szegedy et al., 2016) or ResNet (He et al., 2016) are also available. The purpose of using pre-training is to save the consumption of computing resources and training time. Next, we define the meta-halftone set $\mathcal{M}$, which consists of a set of meta-halftone vectors $\boldsymbol{m}_i (1 \leq i \leq k)$:

$$\mathcal{M} = [\boldsymbol{m}_1, \boldsymbol{m}_2, \boldsymbol{m}_3, ..., \boldsymbol{m}_k], \tag{11}$$

where $k$ represents the number of meta halftones. We obtain $\boldsymbol{m}_i$ by the halftone dithering diffusion states of a set of low-frequency images. Specifically, we construct $k$ images $\mathbf{I}_1, \mathbf{I}_2, \mathbf{I}_3..., \mathbf{I}_k$ with a large range of low-frequency regions ($e.g.$, images with constant gray levels), and obtain the halftones $\mathbf{H}_1, \mathbf{H}_2, \mathbf{H}_3, ..., \mathbf{H}_k$ accordingly using conventional halftone algorithms ($e.g.$, Floyd-Steinberg Algorithm (Lau and Arce, 2018)). We first train for a certain step using the traditional diffusion model alone, where the model uses U-net architecture common in (Ho et al., 2020; Dhariwal and Nichol, 2021) without adding the proposed meta-halftone guided network. We have

$$p_{\theta_1}^{s_1}(\boldsymbol{x}_{t-1}^{\mathbf{H}_i}|\boldsymbol{x}_t^{\mathbf{H}_i}, \boldsymbol{r}^{\mathbf{H}_i}, \boldsymbol{l}^{\mathbf{H}_i}) = \mathcal{N}(\boldsymbol{x}_{t-1}^{\mathbf{H}_i}; \boldsymbol{\mu}_{\theta_1}^{s_1}(\boldsymbol{x}_t^{\mathbf{H}_i}, \boldsymbol{r}^{\mathbf{H}_i}, \boldsymbol{l}^{\mathbf{H}_i}, t), \boldsymbol{\Sigma}_{\theta_1}^{s_1}(\boldsymbol{x}_t^{\mathbf{H}_i}, \boldsymbol{r}^{\mathbf{H}_i}, \boldsymbol{l}^{\mathbf{H}_i}, t)), \tag{12}$$

where the superscript $\mathbf{H}_i$ denotes the constructed halftone sample from $\mathbf{H}_i \in \{\mathbf{H}_1, \mathbf{H}_2, ..., \mathbf{H}_k\}$, and $\boldsymbol{r}^{\mathbf{H}_i}$ and $\boldsymbol{l}^{\mathbf{H}_i}$ can be derived from $\mathbf{I}_i \in \{\mathbf{I}_1, \mathbf{I}_2, ..., \mathbf{I}_k\}$ accordingly. The superscript $s_1$ of $\boldsymbol{x}_{t-1}^{\mathbf{H}_i}, \boldsymbol{x}_t^{\mathbf{H}_i}$ is omitted here for brevity. We can get $\boldsymbol{\epsilon}_{\theta_1}^{s_1}(\sqrt{\bar{\alpha}_t}\boldsymbol{x}_0^{\mathbf{H}_i} + \sqrt{1-\bar{\alpha}_t}\boldsymbol{\epsilon}, t)$ from model predictions and calculate $\boldsymbol{\mu}_{\theta_1}^{s_1}$:

$$\boldsymbol{\mu}_{\theta_1}^{s_1}(\boldsymbol{x}_t^{\mathbf{H}_i}, t) = \frac{1}{\sqrt{\alpha_t}}\Big(\boldsymbol{x}_t^{\mathbf{H}_i} - \frac{\beta_t}{\sqrt{1-\bar{\alpha}_t}}\boldsymbol{\epsilon}_{\theta_1}^{s_1}(\boldsymbol{x}_t^{\mathbf{H}_i}, t)\Big). \tag{13}$$

We use $\boldsymbol{m}_i$ in Equation 11 to represent $\boldsymbol{\mu}_{\theta_1}^{s_1}$ and construct meta-halftone set $\mathcal{M}$ accordingly:

$$\boldsymbol{m}_i = \boldsymbol{\mu}_{\theta_1}^{s_1}(\sqrt{\bar{\alpha}_t}\boldsymbol{x}_0^{\mathbf{H}_i} + \sqrt{1-\bar{\alpha}_t}\boldsymbol{\epsilon}, t), 1 \leq i \leq k. \tag{14}$$

Next, the model learns affine relationships between the hidden feature $\boldsymbol{f}(\boldsymbol{x}_t^{s_1})$ and the meta-halftone set $\mathcal{M}$ through an affine learning layer, and then obtains affine factors $\boldsymbol{g}(\boldsymbol{x}_{t\leftarrow i}^{s_1})$:

$$\boldsymbol{g}(\boldsymbol{x}_{t\leftarrow i}^{s_1}) = \boldsymbol{w}_{g\leftarrow i} \cdot \boldsymbol{f}(\boldsymbol{x}_t^{s_1}) + \boldsymbol{b}_{g\leftarrow i}, \tag{15}$$

where $\boldsymbol{w}_{g\leftarrow i}$ and $\boldsymbol{b}_{g\leftarrow i}$ represent learnable weights and biases, respectively. With the calculated $\boldsymbol{g}(\boldsymbol{x}_{t\leftarrow i}^{s_1})$, we perform a depth-wise aggregation of the meta-halftone set $\mathcal{M}$ to learn the refined representations on depth channels:

$$\widetilde{\boldsymbol{m}} = \sum_{i=1}^{k} \boldsymbol{m}_i \cdot \boldsymbol{g}(\boldsymbol{x}_{t\leftarrow i}^{s_1}). \tag{16}$$

Besides depth information, spatial information is also crucial for meta-halftone guidance. Meta-halftones can provide guidance for the generation of new dithering patterns in local areas. In light of this, we next perform a spatial-wise aggregation of the meta-halftone refined representation $\widetilde{\boldsymbol{m}}$ and the dithering diffusion state vector $\boldsymbol{x}_t^{s_1}$, and denote the result as $\widetilde{\boldsymbol{m}}'$:

$$\widetilde{\boldsymbol{m}}' = \sum \boldsymbol{d} * \big(\widetilde{\boldsymbol{m}} \oplus \boldsymbol{x}_t^{s_1}\big), \tag{17}$$

where $\oplus$ represents the element-wise concatenation in the spatial dimension, $*$ denotes the spatial convolution operation, and $\boldsymbol{d}$ represents the convolution kernel. The output $\boldsymbol{o}$ of the meta-halftone guided network is determined by the meta-halftone guidance $\widetilde{\boldsymbol{m}}'$ and dithering diffusion state vector $\boldsymbol{x}_t^{s_1}$:

$$\boldsymbol{o} = \widetilde{\boldsymbol{m}}'\boldsymbol{x}_t^{s_1}, \tag{18}$$

and $\boldsymbol{o}$ is subsequently fed into U-net as in previous work (Dhariwal and Nichol, 2021; Nichol and Dhariwal, 2021) for model predictions.

## 3.3  Inverse Halftoning Diffusion Conditioned on $x_0^{s_1}$

The goal of inverse halftoning process is to learn a mapping from halftone distributions to continuous-tone distributions and reduce the loss of information in the process. It is similar to halftone dithering diffusion process in Section 3.1, but with different diffusion conditions. For the inverse halftoning diffusion process, we deal with the process $q^{s_2}(x_{1:T'}^{s_2}|x_0^{s_2})$ that gradually adds noise to the data $x_0^{s_2}$ from a continuous-tone distribution $q^{s_2}(x_0^{s_2})$, and the process $q^{s_2}(x_{t-1}^{s_2}|x_t^{s_2})$ of gradually denoising from Gaussian noise to obtain desired samples.

We use a model $p_{\theta_2}^{s_2}(x_{t-1}^{s_2}|x_t^{s_2}, h)$ to estimate $q^{s_2}(x_{t-1}^{s_2}|x_t^{s_2}, x_0^{s_2})$, which is conditioned on the halftone distribution $h$ (i.e., sampling $x_0^{s_1}$ from halftone dithering diffusion model):

$$p_{\theta_2}^{s_2}(x_{0:T'}^{s_2}|h) = p^{s_2}(x_T^{s_2}) \prod_{t=1}^{T'} p_{\theta_2}^{s_2}(x_{t-1}^{s_2}|x_t^{s_2}, h), \tag{19}$$

$$p_{\theta_2}^{s_2}(x_{t-1}^{s_2}|x_t^{s_2}, h) = \mathcal{N}(x_{t-1}^{s_2}; \mu_{\theta_2}^{s_2}(x_t^{s_2}, h, t), \Sigma_{\theta_2}^{s_2}(x_t^{s_2}, h, t)). \tag{20}$$

Pixel-wise guidance between the state $x_t^{s_2}$ and the halftone condition $h$ is applied in the inverse halftoning diffusion process:

$$x_t^{s_2} := x_t^{s_2} \oplus h. \tag{21}$$

where $\oplus$ denotes the concatenate operation. The rest parts (such as the training objective) are similar to the halftone dithering diffusion process, and we omit them here to save space.

## 4  Experiments

### 4.1  Experimental Setup

**Datasets.** In order to evaluate the generalization ability of the proposed model CDH to different halftoning methods, we construct a dataset with relatively strong domain shifts. The domain shifts mainly arise from two aspects: the halftone algorithm and the image semantic. We construct the training set and validation set based on the UTKFace dataset (Zhang et al., 2017) and the test set based on the VOC2012 dataset (Everingham et al., 2010). We collect 9 different halftone dithering algorithms, namely Floyd-Steinberg Dithering, Jarvis-Judice-Ninke Dithering, Stucki Dithering, Atkinson Dithering, Burkes Dithering, Sierra Dithering, and several of their variants (Lau and Arce, 2018). We use halftones generated by 5 of the algorithms in training and validation sets, and halftones generated by remaining 4 algorithms are used as test sets. There are $7,857$ images in the training set, $400$ images in the validation set, and $400$ images in the test set.

**Baselines.** We choose the baselines that are commonly used in inverse halftoning tasks, which include the U-net based methods, generative adversarial network based methods, and super-resolution based methods: **PRL** (Xia and Wong, 2018) proposes an inverse halftoning network with progressively residual learning, which synthesizes the global tone and subtle details from halftone images in a progressive manner. **Dhariwal et al.** (Dhariwal and Nichol, 2021) show that diffusion models can achieve superior image sampling quality than existing generative models, and they use an improved U-net model architecture to achieve high-quality image synthesis. **Nichol et al.** (Nichol and Dhariwal, 2021) show that with some simple modifications, denoising diffusion probabilistic models can also achieve competitive log-likelihood while maintaining high sample quality. They learn variances of reverse diffusion process using reparameterizations and a hybrid learning objective that combines the variational lower-bound with the simplified objective from (Ho et al., 2020). In additon, they improve the noise schedule from a linear noise schedule to a cosine noise schedule. **Song et al.** (Song et al., 2020a) introduces denoising diffusion implicit models, giving rise to implicit models that produce high quality samples much faster. **ESRGAN** (Wang et al., 2018) is used as baselines in previous inverse halftoning work, which introduces the residual-in-residual dense block to generate realistic textures and avoid unpleasant artifacts. **GLEAN** (Chan et al., 2021) is applied to the super-resolution task, using pre-trained generative adversarial networks to solve ill-posed problems in image restoration. We first smooth the halftone images with Gaussian kernels of different sizes, and then use GLEAN for image restoration. **RealESRGAN** (Wang et al., 2021) is extended to a practical image restoration application, which introduces a high-order degradation modeling process to better simulate complex real-world degradations and employs a U-net discriminator with spectral normalization to increase discriminator capability.

Table 1: Model performance comparison of the proposed method CDH and baseline methods in terms of PSNR and SSIM.

| Method | Variants | PSNR | SSIM |
| --- | --- | --- | --- |
| PRL (Xia and Wong, 2018) | – | 22.82 | 0.698 |
| Dhariwal *et al.* (Dhariwal and Nichol, 2021) | – | 23.35 | 0.693 |
| Nichol *et al.* (Nichol and Dhariwal, 2021) | Cosine Noise Schedule | 22.40 | 0.683 |
| Nichol *et al.* (Nichol and Dhariwal, 2021) | Learn Sigma | 22.45 | 0.702 |
| Nichol *et al.* (Nichol and Dhariwal, 2021) | Importance Sampled VLB | 19.99 | 0.615 |
| Song *et al.* (Song et al., 2020a) | DDIM | 18.20 | 0.571 |
| ESRGAN (Wang et al., 2018) | – | 20.20 | 0.428 |
| ESRGAN (Wang et al., 2018) | lkernel | 22.47 | 0.645 |
| GLEAN (Chan et al., 2021) | – | 20.11 | 0.491 |
| GLEAN (Chan et al., 2021) | lkernel | 21.74 | 0.607 |
| RealESRGAN (Wang et al., 2021) | – | 21.04 | 0.626 |
| RealESRGAN (Wang et al., 2021) | lkernel | 20.94 | 0.619 |
| CDH (Ours) | – | **24.24** | **0.727** |

**Evaluation Metrics.** Following previous inverse halftoning work (Xia and Wong, 2018), we use two metrics to evaluate the performance of different models: Peak Signal to Noise Ratio (PSNR) and Structural Similarity (SSIM). PSNR expresses the ratio between the power of peak signals and the power of noises, and SSIM evaluates the similarity between images in terms of luminance, contrast, and structure.

**Implementation Details.** The image size for halftone dithering and inverse halftoning is $256 \times 256$. The channel number of input halftones is $1$. For halftone dithering diffusion model, the learning rate is set to $0.0001$, and $k$ is set to $20$. We use $200$ diffusion steps in training and testing phases. The number of model channels is $64$ and the linear noise schedule is adopted throughout diffusion. We adopt AdamW (Loshchilov and Hutter, 2018) optimizer to train the halftone dithering diffusion model. For inverse halftoning diffusion model, we set the learning rate to $0.0001$ and use $800$ steps in the diffusion process. We set the number of model attention heads to $4$. AdamW (Loshchilov and Hutter, 2018) is also used for optimization.

## 4.2 Experimental Results

We evaluate the performance of the proposed method CDH and baseline methods in terms of PSNR and SSIM, and the experimental results are shown in Table 1. We evaluate the effect of different Gaussian kernel sizes on baseline performance, where ESRGAN (lkernel), GLEAN (lkernel), RealESRGAN (lkernel) denote the use of Gaussian kernels of size 7, and ESRGAN, GLEAN, RealESRGAN represent the use of Gaussian kernels of size 3, respectively. The experimental results show that our method achieves the best performance on both metrics. Previous methods such as PRL (Xia and Wong, 2018), due to the lack of considering the generalization ability of the model on different halftoning algorithms, show sub-optimal performance on test data with relatively strong domain shifts. Compared with traditional diffusion models (Dhariwal and Nichol, 2021; Nichol and Dhariwal, 2021; Song et al., 2020a), the proposed method CDH achieves better results by taking into account the blue noise characteristics in the halftone dithering process, which enables the model to learn from a more realistic and diverse halftone distribution. Different Gaussian kernel sizes affect the performance of image restoration baselines, such as (Wang et al., 2018; Chan et al., 2021; Wang et al., 2021). A larger Gaussian kernel erases dot dithering patterns in halftones to a greater extent, resulting in restored images with less halftone artifacts, but also loss of high frequency information. Our method performs better than the image restoration baselines with different Gaussian kernel sizes. This shows that simply using traditional image restoration approaches is not suitable for the inverse halftoning task, since they do not take into account diverse pixel dithering patterns unique to halftone images.

To verify the effectiveness of the proposed meta-halftone guided network, we conduct experiments to observe the effect on generated halftone dithering qualities with and without the meta-halftone guided network. Experimental results are demonstrated in Figure 4. From the results shown in Figure 4, it can be observed that when the meta-halftone guided network is removed, the generated

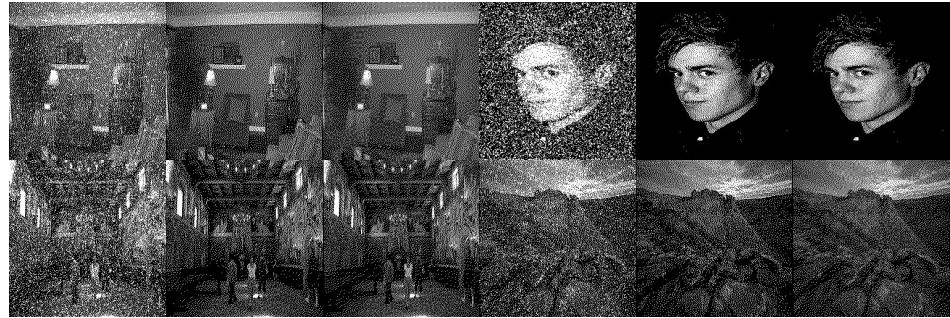

| w/o Meta-halftone Guided Network | w/ Meta-halftone Guided Network | Groundtruth Dithering | w/o Meta-halftone Guided Network | w/ Meta-halftone Guided Network | Groundtruth Dithering |

Figure 4: Experimental results of the meta-halftone guided network. We show the generated halftone dithering results w/ and w/o the proposed meta-halftone guided network, as well as the halftone dithering groundtruth.

Table 2: Model performance comparison on halftones generated by different halftoning algorithms in terms of PSNR and SSIM.

| Halftoning Algorithms | PSNR | SSIM |
|---|---|---|
| Floyd-Steinberg Dithering | 24.46 | 0.735 |
| Simple Floyd-Steinberg Dithering | 24.01 | 0.692 |
| Jarvis-Judice-Ninke Dithering | 24.42 | 0.749 |
| Stucki Dithering | 24.53 | 0.749 |
| Atkinson Dithering | 23.08 | 0.710 |
| Burkes Dithering | 24.69 | 0.746 |
| Sierra Dithering | 24.49 | 0.750 |
| Sierra Lite Dithering | 24.40 | 0.733 |
| Two row Sierra Dithering | 24.54 | 0.741 |

halftones have more obvious dithering artifacts, resulting in poorer visual quality. When using the meta-halftone guided network, the visual quality of generated halftones is improved and more similar to the groundtruth. The experimental results demonstrate the effectiveness of the proposed meta-halftone guided network.

### 4.3   Study the Effect of Different Halftoning Algorithms

In order to explore the performance of CDH on different halftoning algorithms, we conduct experiments on halftones generated by 9 halftoning algorithms, namely Floyd-Steinberg Dithering, Jarvis-Judice-Ninke Dithering, Stucki Dithering, Atkinson Dithering, Burkes Dithering, Sierra Dithering, and several of their variants (Lau and Arce, 2018). The experimental results are shown in Table 2. From the experimental results we observe that the proposed method achieves similar results on different halftoning algorithms, which also verifies good generalization abilities of our method to different halftoning algorithms.

### 4.4   Qualitative Results

To further verify the effectiveness of the proposed method, we also conduct qualitative experiments to compare the performance of our methods and baselines, which are illustrated in Figure 5. From the qualitative experimental results, it can be observed that continuous-tone images generated by baseline methods still have some redundant artifacts, which are dot dithering patterns that are not completely removed during the inverse halftoning process. In contrast, our model removes the artifacts well without introducing redundant patterns, validating the effectiveness of the proposed method.

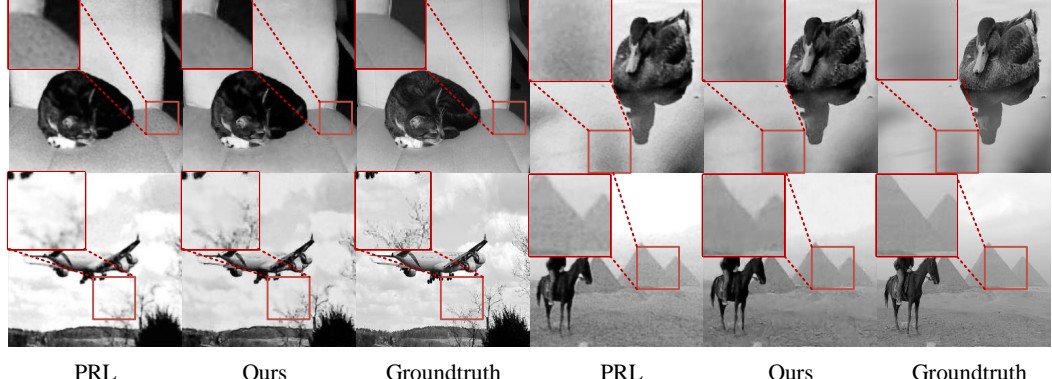

| PRL | Ours | Groundtruth | PRL | Ours | Groundtruth |

Figure 5: Illustration of the qualitative experimental results. We show the inverse halftoning results generated by baseline methods and our method, as well as the groundtruth.

## 5   Conclusion

In this work, we propose a Conditional Diffusion model for image Halftoning (CDH), which consists of a halftone dithering process and an inverse halftoning process. We introduce a generative halftoning method, which regards the black pixels in halftones as physically moving particles, and makes the randomly distributed particles move under some certain guidance through the reverse diffusion process, so as to obtain the desired halftone dithering patterns. A meta-halftone guided network is introduced to avoid redundant patterns and undesired artifacts in generated halftones. By adopting more diverse halftones from halftone dithering diffusion process, we further improve the generalization ability of inverse halftoning model. We conduct experiments on the dataset consisting of 9 halftoning algorithms, and quantitative and qualitative experiments demonstrate the effectiveness of the proposed method.

Acknowledgement: The research is supported by Science and Technology Innovation 2030 - New Generation Artificial Intelligence (2020AAA0104401), Beijing Natural Science Foundation (Z190001), and Peng Cheng Laboratory Key Research Project No.PCL2021A07.

## Broader Impact

A potential negative side effect of this work is that the sample generation model may be used to generate fake images in the halftoning or inverse halftoning process for some certain purpose. In addition, most of the images we used in the dataset are collected from the Internet and may contain some bias. The biases are preserved in the learning of the model and may be reflected in the generated image samples (*e.g.*, induce the model to produce undesired results). An example of this work being used for unethical purposes is to first train the model with biased or discriminatory data and then induce the model to produce unfaithful results when inverse halftoning images. This may misinterpret the original meaning of some ancient prints and mislead people.

## Limitation Analysis

A limitation of our model is that the diffusion process is relatively slow and thus requires more inference time than traditional inverse halftoning models, which is limited when the model is applied to mobile devices. One way to address this limitation is to design parallel inverse halftoning methods, such as restoring low-frequency and high-frequency components of images in parallel, thereby increasing the sampling speed.

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
