# Conditional Diffusion Process for Inverse Halftoning (Supplementary Material)

**Hao Jiang**
Peking University
jianghao@stu.pku.edu.cn

**Yadong Mu**[*]
Peking University
Peng Cheng Laboratory
myd@pku.edu.cn

## 1   Quantitative Experiments of Meta-halftone Guided Network

To verify the effectiveness of the proposed meta-halftone guided network, we conduct quantitative experiments to evaluate the quality of generated halftones. Specifically, we evaluate the generated halftones in terms of tone consistency and structure consistency (Pang et al., 2008; Xia et al., 2021). Tone consistency is calculated by the peak signal-to-noise ratio between the generated halftones and the input continuous-tone images, where the halftones are smoothed by a Gaussian filter kernel. Structure consistency is computed by SSIM metric between the generated halftones and the input continuous-tone images. We compare the halftone quality with and without meta-halftone guided networks in quantitative experiments, and the experimental results are shown in Table 1. It can be observed from Table 1 that when using the meta-halftone guided network, the generated halftones have higher tone consistency and structure consistency with original continuous-tone images, which verifies the effectiveness of the proposed meta-halftone guided network.

Table 1: Quantitative experimental results of generated halftone quality with and without the meta-halftone guided network.

| Method | Tone Consistency | Structure Consistency |
|---|---|---|
| w/o Meta-halftone Guided Network | 32.83 | 0.1253 |
| w/ Meta-halftone Guided Network | **33.39** | **0.1359** |

## 2   More Qualitative Results

To further verify the effectiveness of the proposed method CDH, we conduct more qualitative experiments. The experimental results are shown in Figure 1. We observe that the baseline method (Xia and Wong, 2018) produces inverse halftoning results with more artifacts, while our method removes the artifacts better and thus results in continuous-tone images that are more similar to groundtruth. The qualitative experiments further demonstrate the effectiveness of the proposed method.

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

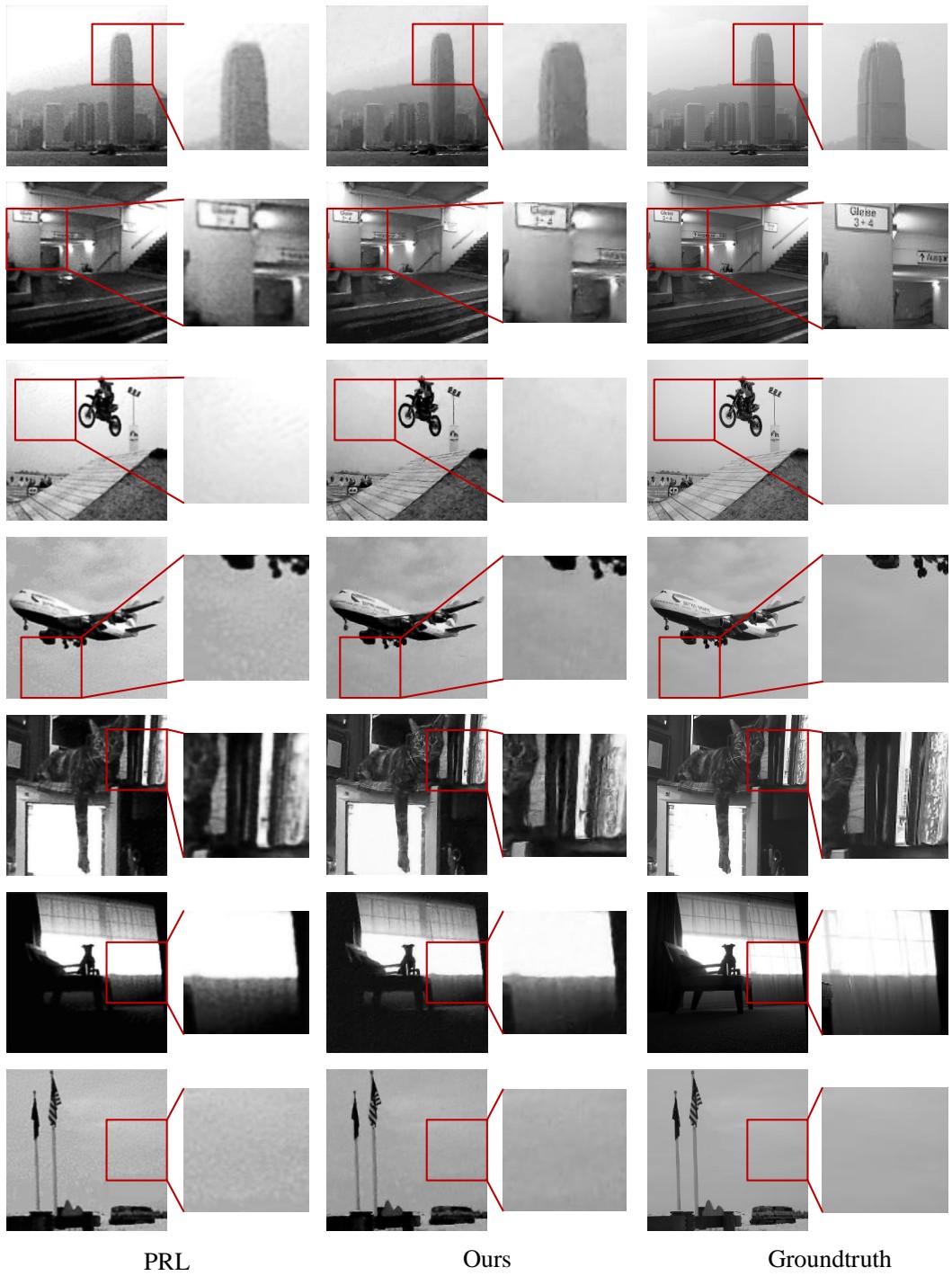

PRL                          Ours                         Groundtruth

Figure 1: Illustration of more qualitative experimental results. We show the inverse halftoning results generated by the baseline method and our method, as well as the groundtruth.