# OpenReview forum: "Conditional Diffusion Process for Inverse Halftoning"
_NeurIPS.cc/2022/Conference — NeurIPS 2022 Accept_

### Official Review · Reviewer_6UuY · 2022-07-03

**Rating:** 5
**Confidence:** 2
**Soundness:** 2 fair
**Presentation:** 3 good
**Contribution:** 2 fair

**Summary:**

This paper introduces an inverse halftoning method based on conditional diffusion processes. The goal is to restore halftoned images generated using any halftoning algorithm, by leveraging recent advances on generative models. Previous work did not generalize properly to halftoning algorithms outside of those used for training the model, while the proposed method performs the inverse halftoning operation through conditional diffusion models. To improve the capabilities of the model, the paper ontroduces a meta-halftone guiding framework for introducing blue noise into the diffusion process. Quantitative results are provided that show improved results with respect to supervised learning and image filtering frameworks.

**Questions:**

- What is the impact of the design of the meta-halftone module? Why is this design chosen? Could you provide a baseline for this design and some ablations on why this design is better than other methods? I am concerned about the use of a pretraiend VGG network for this purpose, as the features on its last layer may be too closely related to ImageNet classification. Why is pre-training needed?
- What does it mean that the "mainstream U-net architecture has poor generalization ability in practical applications"? Is it a problem of the U-net architecture by itself? If so, what does this paper do to mitigate these problems? Looking at the code, it seems that the diffusion model is a U-net with attention, similar to those used in Palette or other DDPM models. Can the authors provide more details on this? Is it really an architectural problem, or are more factors involved (learning formulation, loss function, etc.)?
- On the validation side, I think PSNR and SSIM are not sufficient for analyzing the results of generative models. What are the scores using perceptual metrics (FiD, Inception Score, LPIPS)?
- On Table 1, why are standard deviations only provided for CDH? Why is the standard deviation for SSIM so low?
- What happens with RGB images? Does this method only work for grayscale images?
- On Figure 1 and Figure 5, what is the input to the model? It would be interesting to see the halftoned inputs used throughout the paper.
- Does the model work better with some halftoning algorithms than others? A quantitative analysis on this would be benefitial.
- Could the authors compare this paper to Palette: Image-to-Image Diffusion Models, Saharia et al. and Deep Image Prior, Ulyanov et al. ?  I think the image restoration ideas on these papers are relevant for this CDH paper and should be at least mentioned.
- What happens when the images are larger to the 256x256 pixels used in training?
- Can the authors provide an illustration on the "blue noise" that can be introduced by the proposed metahalftone module?

**Limitations:**

The authors have adressed some limitations of the proposed method, particularly issues of computational cost at inference time with respect to other work. They also point out that their work may be used to "generate fake images for some certain purpose", and that the dataset used for training "may contain some bias".

I suggest the authors extend this analysis. In particular, I think it would be interesting to give particular examples of how this work may be used for unethical purposes, or which harmful biases are introduced with the dataset. Also, I suggest the authors to extend the limitation analysis, in terms of:
- What happens with RGB images?
- What happens with resolutions outside of the specified 256x256 pixels?

**Strengths And Weaknesses:**

*Summary*

This paper provides valuable ideas for designing diffusion models for inverse halftoning. The paper is in general well written, but lacks sufficient validation, appropriate discussion of previous work and the significance of the findings of this paper are limited in the overall literature of diffusion models.


Strenghts:
- This paper provides specific ideas on how to leverage diffusion models for the particular problem of inverse halftoning.
- The paper is, in general, well written and is very easy to follow. The authors present their ideas in a concise manner.
- Code is provided for enhanced reproducibility.
- Provided analysis show significant improvements with respect to previous work.


Weaknesses:
- The paper lacks sufficient validation, it is not clear why some decisions were taken. Please see "questions" for details.
- The design of the Meta-halftone guided network is not adequately justified. Quantitatevely, it is shown that this module enhances the result of the model, but it is not clear to the reader why this design was chosen.
- Previous work is not adequately studied. I suggest the authors to consider analyzing Palette: Image-to-Image Diffusion Models, Saharia et al. and Deep Image Prior, Ulyanov et al.
- The analysis of previous work is also limited. Following the authors wording, it is claimed that "U-net architecture has poor generalization in practical applications". Reading this, it may be understood that this paper provides alternatives to the U-net architecture, but it does not. I suggest the authors to change the wording to "supervised learning" or "pixel-wise regression" instead of "mainstream U-net architecture", as this represents more accurately the baselines that the authors want to improve.
- Limitations need further analysis.

---

> ### Author Response · Authors · 2022-08-02
> **Author Responses to Reviewer 4 (6UuY) (1/3)**
>
> Dear Reviewer:
> Thank you very much for your constructive comments on our
> paper! Our responses to your questions are below.
>
> _Q1: The impact of meta-halftone guided network and its motivation._
>
> The motivation for designing the meta-halftone guided network is to introduce good blue noise properties during halftone dithering. Blue noise is essential for generating high-quality halftones, which avoids noticeable low frequency visual artifacts in the generated halftones by forcing random pixel dithering. To achieve this goal, the meta-halftone guided network constructs a meta-halftone set with the help of the diffusion state vectors of $k$ constant grayscale images, and guides the generation of new halftone dithering patterns through these meta-halftone vectors.
>
> To verify the effectiveness of the proposed meta-halftone guided network, we conduct ablation experiments to evaluate the quality of generated halftones. Specifically, we evaluate the generated halftones in terms of tone consistency and structure consistency (Pang et al., 2008; Xia et al., 2021). Tone consistency is calculated by the peak signal-to-noise ratio between the generated halftones and the input continuous-tone images, where the halftones are smoothed by a Gaussian filter kernel. Structure consistency is computed by SSIM metric between the generated halftones and the input continuous-tone images. We compare the halftone quality with and without meta-halftone guided networks in quantitative experiments, and the experimental results are shown below.
>
> Method | Structure Consistency | Tone Consistency
> - | - | -
> Meta-halftone Guided Network | 0.1283 | 24.51
> Meta-halftone Guided Network | 0.1550 | 26.56
>
> It can be observed that when using the meta-halftone guided network, the generated halftones have higher tone consistency and structure consistency with original continuous-tone images, which verifies the effectiveness of the proposed meta-halftone guided network.
>
> The pretrained VGG network is used for feature extraction of constant grayscale images, and we also perform feature mapping (Eq. 15) on the VGG features to learn task-related feature representations. The reason for using pre-training is to save the consumption of computing resources and training time, making the meta-halftone guided network more lightweight (only contains 81K training parameters). Thank you for your suggestion, we will further explore the performance of using the learnable VGG network and add a comparison of the results in the revised version of the paper.
>
>
> _Q2: Inaccurate wording of "U-net architecture"._
>
> Thank you for your valuable suggestion, we will clarify this in the paper and revise "U-net architecture" to "supervised learning with element-wise regression".
>
>
> _Q3: Evaluation scores using perception metrics._
>
> According to the properties of the halftone task, we evaluate the performance of our model on the perceptual similarity metric (LPIPS) and compare it with baseline methods. The results are as follows:
>
> Method | Variants | LPIPS
> -|-|-
> (Wang et al., 2018) | ESRGAN | 0.494
> (Wang et al., 2018) | ESRGAN lkernel | 0.406
> (Chan et al., 2021) | GLEAN | 0.377
> (Chan et al., 2021) | GLEAN lkernel | 0.233
> (Lee et al., 2022) | AP-BSN DND | 0.664
> (Lee et al., 2022) | AP-BSN SIDD | 0.512
> (Lee et al., 2022) | AP-BSN SIDD-ben | 0.486
> (Lee et al., 2022) | AP-BSN NIND | 0.576
> (Dhariwal et al., 2021) | DDPM, Channel 64, Res 1 | 0.208
> (Dhariwal et al., 2021) | DDPM, Channel 64, Res 3 | 0.224
> (Dhariwal et al., 2021) | DDPM, Channel 128, Res 1 | 0.278
> (Dhariwal et al., 2021) | DDPM, Channel 128, Res 2 | 0.222
> (Dhariwal et al., 2021) | DDPM, Channel 128, Res 3 | 0.230
> CDH (Ours) | --- | __0.198__
>
>
> A lower LPIPS score indicates a greater similarity between the generated halftone and continuous-tone groundtruth, and it can be observed that our proposed method achieves the best results among all baselines.
>
> _Q4: Standard deviations on Table 1._
>
> Thanks for your suggestion and we will add the standard deviation of baseline models in the revised version. The reason for the low standard deviation of SSIM may be that SSIM focuses on the structural similarity of images, and the variation is smaller than that of PSNR.
>
> _Q5: Inverse halftoning on RGB images._
>
> Our method can also work on RGB images, and the model performance on RGB images is as follows:
>
> Input Halftone | PSNR | SSIM
> -|-|-
> RGB images | 26.20 | 0.853
>
> _Q6: Inputs to the model on Figure 1 and Figure 5._
>
> Yes, the inputs to the model on Figure 1 and Figure 5 are both halftone images.

---

> ### Author Response · Authors · 2022-08-02
> **Author Responses to Reviewer 4 (6UuY) (2/3)**
>
> _Q7: Does the model work better with some halftoning algorithms than others?_
>
> In order to explore the performance of the model on different halftoning algorithms, we conduct experiments on images generated by 9 halftoning algorithms, namely Floyd-Steinberg Dithering, Jarvis-Judice-Ninke Dithering, Stucki Dithering, Atkinson Dithering, Burkes Dithering, Sierra Dithering, and several of their variants (Lau and Arce, 2018). The experimental results are as follows:
>
> Method | PSNR | SSIM
> -|-|-
> Floyd-Steinberg | 24.46 | 0.735
> Simple Floyd-Steinberg | 24.01 | 0.692
> Jarvis-Judice-Ninke | 24.42 | 0.749
> Stucki | 24.53 | 0.749
> Atkinson | 23.08 | 0.710
> Burkes | 24.69 | 0.746
> Sierra | 24.49 | 0.750
> Sierra Lite | 24.40 | 0.733
> Two row Sierra | 24.54 | 0.741
>
>
> We can observe that the proposed method achieves similar results on different halftoning algorithms, which also verifies the good robustness of our method.
>
> _Q8: Compare with more related methods._
>
> For a more comprehensive comparison, we add some methods proposed for diffusion models (Dhariwal et al., 2021; Quinn et al., 2021; Song et al., 2021), image denoising (Lee et al., 2022), and deep image priors (Ulyanov et al., 2018) as baselines, and compare the performance of our method with these baselines.
>
>
> Method | Variants | PSNR | SSIM
> -|-|-|-
> (Dhariwal et al., 2021) | Channel 64, Res 1 | 22.46 | 0.670
> (Dhariwal et al., 2021) | Channel 64, Res 2 | 21.97 | 0.638
> (Dhariwal et al., 2021) | Channel 64, Res 3 | 22.97 | 0.690
> (Dhariwal et al., 2021) | Channel 128, Res 1 | 23.21 | 0.694
> (Dhariwal et al., 2021) | Channel 128, Res 2 | 22.12 | 0.634
> (Dhariwal et al., 2021) | Channel 128, Res 3 | 23.35 | 0.693
> (Quinn et al., 2021) | Cosine noise schedule | 22.40 | 0.683
> (Quinn et al., 2021) | Learn sigma | 22.45 | 0.702
> (Quinn et al., 2021) | Importance sampled VLB | 19.99 | 0.615
> (Song et al., 2021) | DDIM | 18.20 | 0.571
> (Lee et al., 2022) | AP-BSN DND | 15.84  | 0.512
> (Lee et al., 2022) | AP-BSN SIDD | 20.89 | 0.536
> (Lee et al., 2022) | AP-BSN SIDD-ben | 17.81 | 0.565
> (Lee et al., 2022) | AP-BSN NIND | 13.93 | 0.333
> (Ulyanov et al., 2018) | --- | 9.31 | 0.568
> CDH (Ours) | --- | __24.24__ | __0.727__
>
> Compared with traditional diffusion models (Dhariwal et al., 2021; Quinn et al., 2021; Song et al., 2021), the proposed method CDH achieves better results by taking into account the blue noise characteristics in the halftone dithering process, which enables the model to learn from a more realistic and diverse halftone distribution. Our model also achieves better results compared to image denoising methods (Lee et al., 2022) and deep image prior based methods (Ulyanov et al., 2018). This shows that simply using traditional image restoration approaches is not suitable for the inverse halftoning task, since they do not take into account the diverse pixel dithering patterns unique to halftone images.
>
>
> _Q9: Inverse halftoning of images larger to the 256$\times$256 pixels._
>
> The results achieved by the method on images with resolutions 512x512 and 1024x1024 are as follows:
>
> Image Resolution | PSNR | SSIM
> -|-|-
> 512x512 | 20.79 | 0.760
> 1024x1024 | 20.62 | 0.731
>
> It can be observed that our method also achieves good results on images with resolutions larger than 256x256.
>
> We also provide the performance comparison of our proposed method and diffusion-based baseline methods when the image resolution is 512x512, as shown below. It can be observed that our method outperforms the baseline methods.
>
>
> Method | Variants (512x512) | PSNR | SSIM
> -|-|-|-
> (Dhariwal et al., 2021) | Channel 64, Res 1 | 17.65 | 0.698
> (Dhariwal et al., 2021) | Channel 64, Res 2 | 18.77 | 0.679
> (Dhariwal et al., 2021) | Channel 64, Res 3 | 17.99 | 0.697
> (Quinn et al., 2021) | Cosine noise schedule | 17.52 | 0.706
> (Quinn et al., 2021) | Learn sigma | 19.24 | 0.705
> (Quinn et al., 2021) | Importance sampled VLB | 17.36 | 0.669
> CDH (Ours) | --- | __20.79__ | __0.760__
>
> _Q10: An illustration on the "blue noise"._
>
> We have added an illustration of blue noise to Figure 4 (right) in the revised paper. It can be observed that the halftone images generated by the proposed meta-halftone guided network have more random dot dithering patterns, which validates the effectiveness of the proposed method.
>
> _Q11: Extend the limitation analysis._
>
> Harmful biases that may be introduced in the dataset, including race, skin color, geography, etc., may contaminate the dataset and induce the model to produce undesired results. An example of this work being used for unethical purposes is to first train the model with biased or discriminatory data and then induce the model to produce unfaithful results when inverse halftoning images. This may misinterpret the original meaning of some ancient prints and mislead people. Thank you for your valuable suggestion, we will add more examples for limitation analysis in the revised version of the paper.
>
>
> Thanks again for your constructive comments!

---

> ### Author Response · Authors · 2022-08-02
> **Author Responses to Reviewer 4 (6UuY) (3/3)**
>
> The references used in our responses.
>
> [1] Menghan Xia, Wenbo Hu, Xueting Liu, and Tien-Tsin Wong. Deep halftoning with reversible binary pattern. ICCV, 2021.
>
> [2] Wai-Man Pang, Yingge Qu, Tien-Tsin Wong, Daniel Cohen-Or, and Pheng-Ann Heng. Structure-aware halftoning. SIGGRAPH, 2008.
>
> [3] Daniel L Lau and Gonzalo R Arce. Modern digital halftoning. CRC Press, 2018.
>
> [4] Wooseok Lee, Sanghyun Son and Kyoung Mu Lee. AP-BSN: Self-Supervised Denoising for Real-World Images via Asymmetric PD and Blind-Spot Network. CVPR, 2022.
>
> [5] Dmitry Ulyanov, Andrea Vedaldi, and Victor Lempitsky. Deep image prior. CVPR, 2018.
>
> [6] Prafulla Dhariwal, and Alexander Nichol. Diffusion models beat gans on image synthesis. NeurIPS, 2021.
>
> [7] Alexander Quinn Nichol, and Prafulla Dhariwal. Improved denoising diffusion probabilistic models. ICML, 2021.
>
> [8] Jiaming Song, Chenlin Meng, and Stefano Ermon. Denoising diffusion implicit models. ICLR, 2021.
>
> [9] Xintao Wang, Ke Yu, Shixiang Wu, Jinjin Gu, Yihao Liu, Chao Dong, Yu Qiao, and Chen Change Loy. Esrgan:
> 383 Enhanced super-resolution generative adversarial networks. ECCV, 2018.
>
> [10] Xintao Wang, Liangbin Xie, Chao Dong, and Ying Shan. Real-esrgan: Training real-world blind super-resolution
> 386 with pure synthetic data. ICCV, 2021.
>
> [11] Robert A Ulichney. Void-and-cluster method for dither array generation. In Human Vision, Visual Processing, and Digital Display, 1993.

---

> ### Comment · Reviewer_6UuY · 2022-08-07
> **Post-rebuttal comments**
>
> The authors have performed a thorough evaluation based on my and other reviewers suggestions. Many of my concerns regarding validation have been addressed adequately. I appreciate the efforts done by the authors during the rebuttal period, and I think that if these new results are included in the final version of the paper or its supplementary material, this paper will be ready for publication. I am therefore willing to increase my original rating.
>
> However, I would still suggest the authors to provide an analysis on computational cost with respect to other methods. Even if their method is worse in comparison, being more transparent on its limitations would not hinder the quality of the paper.

---

> ### Comment · Reviewer_6UuY · 2022-08-08
> **Post-Rebuttal Rating**
>
> After reading the authors response, and the other reviews, I increase my rating to a borderline accept.

---

> ### Author Response · Authors · 2022-08-09
> **Author Responses to Reviewer 4 for Post-rebuttal comments**
>
> Dear reviewer:
>
> Thank you so much for your response and your valuable suggestions!
>
> Sorry for the late reply due to adding some related experiments yesterday. Sincerely hope for your understanding :)
>
> We compare the computational cost of our method and baseline methods, and the results are as follows:
>
>
>
> Method | Variants | GFlops | Params/M
> -|-|-|-
> (Xia and Wong, 2018) | --- | 108.9 | 3.2
> (Wang et al., 2018) | --- | 1174.9 | 16.7
> (Ulyanov et al., 2018) | --- | 20.0 | 2.2
> (Dhariwal et al., 2021) | Res 1 | 150.7 | 24.2
> (Dhariwal et al., 2021) | Res 2 | 171.6 | 34.3
> (Dhariwal et al., 2021) | Res 3 | 192.4 | 44.3
> (Chitwan et al., 2021) | Res 1 | 155.1 | 45.3
> (Chitwan et al., 2021) | Res 2 | 205.6 | 62.6
> (Chitwan et al., 2021) | Res 3 | 256.0 | 80.0
> (Lee et al., 2022) | --- | 1888.3 | 3.7
> CDH (Ours) | --- | 119.2 | 8.5
>
>
> It can be observed that our method has fewer GFlops than traditional image restoration methods (Wang et al., 2018; Lee et al., 2022), but more parameters than (Xia and Wong, 2018; Ulyanov et al., 2018). The reason is that we use a more complex network structure in the diffusion model than the baseline, and thus have slightly more parameters. Compared to diffusion methods (Dhariwal et al., 2021; Chitwan et al., 2021), our method has fewer Gflops and parameters. This is due to the fact that we use halftone-specific structures (_e.g._, meta-halftone-guided network) to replace some layers in traditional diffusion models, resulting in fewer Gflops and parameters. By the way, the proposed meta-halftone guided network has the advantage of being very lightweight, containing only 81K training parameters.
>
> In addition, as a complement to the performance comparison, we also verify the performance of Palette and its variants (Chitwan et al., 2021) as baselines, and the results are as follows.
>
> Method | Variants | PSNR | SSIM
> -|-|-|-
> (Chitwan et al., 2021) | Channel 16, Res 1 | 21.12 | 0.693
> (Chitwan et al., 2021) | Channel 16, Res 2 | 18.15 | 0.652
> (Chitwan et al., 2021) | Channel 16, Res 3 | 19.86 | 0.678
> (Chitwan et al., 2021) | Channel 32, Res 1 | 19.49 | 0.671
> (Chitwan et al., 2021) | Channel 32, Res 2 | 19.00 | 0.667
> (Chitwan et al., 2021) | Channel 32, Res 3 | 19.33 | 0.660
> (Chitwan et al., 2021) | Channel 64, Res 1 | 17.99 | 0.670
> (Chitwan et al., 2021) | Channel 64, Res 2 | 21.10 | 0.690
> (Chitwan et al., 2021) | Channel 64, Res 3 | 20.10 | 0.687
> CDH (Ours) | --- | __24.24__ | __0.727__
>
>
> It can be observed that our method achieves better performance than the baseline methods, validating the effectiveness of the proposed method.
>
> We greatly appreciate your replies to us, and sincerely appreciate all your valuable suggestions, which make our paper more comprehensive. Hope our response answered the question you were looking for. Thanks a lot!
>
>
> [1] Wooseok Lee, Sanghyun Son and Kyoung Mu Lee. AP-BSN: Self-Supervised Denoising for Real-World Images via Asymmetric PD and Blind-Spot Network. CVPR, 2022.
>
> [2] Xintao Wang, Liangbin Xie, Chao Dong, and Ying Shan. Real-esrgan: Training real-world blind super-resolution
> 386 with pure synthetic data. ICCV, 2021.
>
> [3] Prafulla Dhariwal, and Alexander Nichol. Diffusion models beat gans on image synthesis. NeurIPS, 2021.
>
> [4] Saharia Chitwan, Chan William, Chang Huiwen, et al. Palette: Image-to-image diffusion models. arXiv preprint, 2021.
>
> [5] Menghan Xia and Tien-Tsin Wong. Deep inverse halftoning via progressively residual learning. ACCV, 2018.
>
> [6] Xintao Wang, Ke Yu, Shixiang Wu, Jinjin Gu, Yihao Liu, Chao Dong, Yu Qiao, and Chen Change Loy. Esrgan: Enhanced super-resolution generative adversarial networks. ECCV, 2018.
>
> [7] Dmitry Ulyanov, Andrea Vedaldi, and Victor Lempitsky. Deep image prior. CVPR, 2018.

---

### Official Review · Reviewer_Ptjr · 2022-07-07

**Rating:** 3
**Confidence:** 1
**Soundness:** 1 poor
**Presentation:** 1 poor
**Contribution:** 2 fair

**Summary:**

In this paper, the authors propose a generative halftoning method, which regards the black pixels in halftones as physically moving
particles, and makes the randomly distributed particles move under some certain guidance through the reverse diffusion process. The proposed Conditional Diffusion model for image  Halftoning (CDH)  consists of a halftone dithering process and an inverse halftoning process. Specifically, to avoid introducing redundant patterns and undesired artifacts during halftone generation, they propose
a meta-halftone guided network to incorporate the blue noise guidance into the diffusion process and train an inverse halftoning diffusion model to learn the mapping function from the halftone distribution to the continuous-tone distribution. They claim that this helps the model to learn a  more robust mapping to perform inverse halftoning and improve the generalization ability to unseen samples.


**Questions:**

The authors claimed that the limitation of the existing deep learning approaches for inverse halftoning is the lack of generalization due to the use of U-Net.  However, to my understanding, the particular architecture of the U-Net is not the main culprit, but the way of supervised training is the problem. The authors should clarify this.

**Ethics Review Area:**

["I don’t know"]

**Limitations:**

The slow speed of the diffusion model is stated as the limitation.

**Strengths And Weaknesses:**

Strength: This is the first attempt to use diffusion models for the half-tone generation and inverse half-toning.

Weakness:  The writing is quite confusing. The reviewer has tried to understand the link between the proposed method and the score-based diffusion model, but could not find any close link.  Why does the complicated forward diffusion process be necessary? In figure 2, why there are two paths for half-tone dithering and inverse half-toning? If I understand correctly,  a better way of inverse halftoning using diffusion would be a noising process of the dithered image followed by the reverse diffusion process to obtain the continuous image. But the main focus of this paper appears to be the dither pattern generation, which appears quite far distance from the diffusion model. I have to say that I am not an expert on half-toning, but still, the way of describing the algorithm is quite confusing to general NeuRIPS readers.

---

> ### Author Response · Authors · 2022-08-02
> **Author Responses to Reviewer 3 (Ptjr)**
>
> Dear Reviewer:
> Thanks for your constructive comments and sorry for your misunderstanding of our paper. Our responses to your questions are below.
>
> _Q1: Why are there two paths in Figure 2?_
>
> Our main observation is that the existing halftoning methods have poor generalization ability (when the halftone dithering patterns of the training data and test data have a large gap, the inverse halftoning results will have obvious artifacts). To address this issue, we propose a halftone dithering diffusion process equipped with a meta-halftone guided module (Section 3.1, 3.2) to generate diverse and realistic halftone dithering images (the upper path in Figure 2). With more diverse halftone dithering patterns, we propose the inverse halftoning diffusion procedure to learn the mapping from halftone distributions to continuous-tone distributions (the lower path of Figure 2). In other words, the upper path (halftone dithering diffusion) in Figure 2 is used to generate more diverse halftones (used to improve the generalization of the model to different dithering patterns), and these generated results are used as the input of the lower path (inverse halftoning diffusion).
>
> Further, to verify the effectiveness of the upper path in Figure 2, we provide some ablation experiments. We remove the halftone dithering diffusion (the upper path in Figure 2) and compare our method with several traditional diffusion models (Dhariwal et al., 2021; Quinn et al., 2021; Song et al., 2021), and the results are as follows:
>
>
> Method | Variants | PSNR | SSIM
> -|-|-|-
> (Quinn et al., 2021) | Cosine noise schedule | 22.40 | 0.683
> (Quinn et al., 2021) | Learn sigma | 22.45 | 0.702
> (Quinn et al., 2021) | Importance sampled VLB | 19.99 | 0.615
> (Song et al., 2021) | DDIM | 18.20 | 0.571
> (Dhariwal et al., 2021) | Channel 64, Res 1 | 22.46 | 0.670
> (Dhariwal et al., 2021) | Channel 64, Res 2 | 21.97 | 0.638
> (Dhariwal et al., 2021) | Channel 64, Res 3 | 22.97 | 0.690
> (Dhariwal et al., 2021) | Channel 128, Res 1 | 23.21 | 0.694
> (Dhariwal et al., 2021) | Channel 128, Res 2 | 22.12 | 0.634
> (Dhariwal et al., 2021) | Channel 128, Res 3 | 23.35 | 0.693
> CDH (Ours) | --- | __24.24__ | __0.727__
>
>
> Compared with traditional diffusion models (Dhariwal et al., 2021; Quinn et al., 2021; Song et al., 2021), our proposed method CDH performs better, illustrating the effectiveness of our proposed halftone dithering diffusion path in Figure 2. By taking into account the blue-noise properties during halftone dithering, the model is enabled to learn from more realistic and diverse halftone distributions.
>
>
> In addition, to verify the effectiveness of the proposed meta-halftone guided network in the halftone dithering diffusion path, we conduct ablation experiments to evaluate the quality of generated halftones in terms of tone consistency and structure consistency (Pang et al., 2008; Xia et al., 2021). Tone consistency is calculated by the peak signal-to-noise ratio between the generated halftones and the input continuous-tone images, where the halftones are smoothed by a Gaussian filter kernel. Structure consistency is computed by SSIM metric between the generated halftones and the input continuous-tone images. We compare the halftone quality with and without meta-halftone guided networks in quantitative experiments, and the experimental results are as follows:
>
> Method | Structure Consistency | Tone Consistency
> -|-|-
> w/o Meta-halftone Guided Network | 0.1283 | 24.51
> w/ Meta-halftone Guided Network | 0.1550 | 26.56
>
> It can be observed that when using the meta-halftone guided network, the generated halftones have higher tone consistency and structure consistency with original continuous-tone images, which verifies the effectiveness of the proposed meta-halftone guided network.
>
>
> _Q2: Inaccurate wording of "U-net architecture"_
>
> Thank you for your valuable suggestion, we will clarify this in the paper and revise "U-net architecture" to "supervised learning with element-wise regression".
>
> Thanks again for your constructive comments!
>
>
> [1] Prafulla Dhariwal, and Alexander Nichol. Diffusion models beat gans on image synthesis. NeurIPS, 2021.
>
> [2] Alexander Quinn Nichol, and Prafulla Dhariwal. Improved denoising diffusion probabilistic models. ICML, 2021.
>
> [3] Jiaming Song, Chenlin Meng, and Stefano Ermon. Denoising diffusion implicit models. ICLR, 2021.
>
> [4] Wai-Man Pang, Yingge Qu, Tien-Tsin Wong, Daniel Cohen-Or, and Pheng-Ann Heng. Structure-aware halftoning. SIGGRAPH, 2008.
>
> [5] Menghan Xia, Wenbo Hu, Xueting Liu, and Tien-Tsin Wong. Deep halftoning with reversible binary pattern. ICCV, 2021.
>
> [6] Robert A Ulichney. Void-and-cluster method for dither array generation. In Human Vision, Visual Processing, and Digital Display, 1993.

---

> > ### Comment · Reviewer_Ptjr · 2022-08-06
> > **Thanks for your response**
> >
> > Thanks for your response and detailed explanation of why there are two paths. That being said, the clarity of the paper is still a problem, so I would like to keep my original rating.

---

### Official Review · Reviewer_FAj2 · 2022-07-11

**Rating:** 5
**Confidence:** 3
**Soundness:** 3 good
**Presentation:** 3 good
**Contribution:** 3 good

**Summary:**

This paper proposes a new method for inverse halftoning (recovering realistic images from ancient prints). The proposed method is a generative method that uses reverse diffusion process to produce desired halftone dithering patterns. A meta-halftone guided network is proposed to use the blue noise to guide the halftone diffusion process for handling redundant patterns and artifacts. The paper also proposes to condition the inverse halftoning diffusion on the initial state of halftone dithering diffusion, in order to learn a more robust mapping. Experiments demonstrate its state-of-the-art performance.

**Questions:**

Please see above section.

**Limitations:**

Yes.

**Strengths And Weaknesses:**

Strengths.

1. A generative halftoning method is proposed.
2. A meta-halftone guided network is proposed to reduce redundant patterns and undesired artifacts.
3. A learning dataset is constructed.
4. The motivation of this work is clear.
5. Code is provided.

Weaknesses.
1. The split of the proposed dataset (7857 for training, 400 for validation and 400 for test) seems not reasonable.
2. The paper compres to only four methods chosen/adapted from previous halftoning/super-resolution methods, which is not convincing. For example, diffusion models that are proposed for image denoising can be used for comparisons.

---

> ### Author Response · Authors · 2022-08-02
> **Author Responses to Reviewer 2 (FAj2)**
>
> Dear Reviewer:
> Thank you very much for your constructive comments on our paper! Our responses to your questions are below.
>
>
> _Q1: The split of the proposed dataset._
>
> There are a total of 8,657 images in our dataset (each halftone dithered image has a corresponding continuous-tone image). We randomly divided around 10\% of the images in the dataset as validation and test sets (400 images each, non-overlapping each other), and the remaining images were used as training sets (7,857 images). Thank you for your valuable suggestion, we will add the reason for the partitioning of the dataset in the revised version of the paper.
>
> _Q2: Compare more related methods._
>
> For a more comprehensive comparison, we add some methods proposed for diffusion models (Dhariwal et al., 2021; Quinn et al., 2021; Song et al., 2021), image denoising (Lee et al., 2022), and deep image priors (Ulyanov et al., 2018) as baselines, and compare the performance of our method with these baselines. The results are shown below.
>
>
> Method | Variants | PSNR | SSIM
> -|-|-|-
> (Dhariwal et al., 2021) | Channel 64, Res 1 | 22.46 | 0.670
> (Dhariwal et al., 2021) | Channel 64, Res 2 | 21.97 | 0.638
> (Dhariwal et al., 2021) | Channel 64, Res 3 | 22.97 | 0.690
> (Dhariwal et al., 2021) | Channel 128, Res 1 | 23.21 | 0.694
> (Dhariwal et al., 2021) | Channel 128, Res 2 | 22.12 | 0.634
> (Dhariwal et al., 2021) | Channel 128, Res 3 | 23.35 | 0.693
> (Quinn et al., 2021) | Cosine noise schedule | 22.40 | 0.683
> (Quinn et al., 2021) | Learn sigma | 22.45 | 0.702
> (Quinn et al., 2021) | Importance sampled VLB | 19.99 | 0.615
> (Song et al., 2021) | DDIM | 18.20 | 0.571
> (Lee et al., 2022) | AP-BSN DND | 15.84  | 0.512
> (Lee et al., 2022) | AP-BSN SIDD | 20.89 | 0.536
> (Lee et al., 2022) | AP-BSN SIDD-ben | 17.81 | 0.565
> (Lee et al., 2022) | AP-BSN NIND | 13.93 | 0.333
> (Ulyanov et al., 2018) | --- | 9.31 | 0.568
> CDH (Ours) | --- | __24.24__ | __0.727__
>
>
> Compared with traditional diffusion models (Dhariwal et al., 2021; Quinn et al., 2021; Song et al., 2021), our proposed method CDH achieves better results by taking into account the blue noise characteristics in the halftone dithering process, which enables the model to learn from a more realistic and diverse halftone distribution. Our model also achieves better results compared to image denoising methods (Lee et al., 2022) and deep image prior based methods (Ulyanov et al., 2018). This shows that simply using traditional image restoration approaches is not suitable for the inverse halftoning task, since they do not take into account the diverse pixel dithering patterns unique to halftone images.
>
> Thanks again for your constructive comments!
>
>
>
> [1] Wooseok Lee, Sanghyun Son and Kyoung Mu Lee. AP-BSN: Self-Supervised Denoising for Real-World Images via Asymmetric PD and Blind-Spot Network. CVPR, 2022.
>
> [2] Dmitry Ulyanov, Andrea Vedaldi, and Victor Lempitsky. Deep image prior. CVPR, 2018.
>
> [3] Prafulla Dhariwal, and Alexander Nichol. Diffusion models beat gans on image synthesis. NeurIPS, 2021.
>
> [4] Alexander Quinn Nichol, and Prafulla Dhariwal. Improved denoising diffusion probabilistic models. ICML, 2021.
>
> [5] Jiaming Song, Chenlin Meng, and Stefano Ermon. Denoising diffusion implicit models. ICLR, 2021.
>
> [6] Saharia Chitwan, Chan William, Chang Huiwen, et al. Palette: Image-to-image diffusion models. arXiv preprint, 2021.

---

> ### Author Response · Authors · 2022-08-09
> **Author Supplementary Responses to Reviewer 2 (FAj2)**
>
>
> Dear reviewer:
>
> In addition to the added comparisons above, we also verify the performance of Palette and its variants (Chitwan et al., 2021) as baselines, and the results are as follows.
>
> Method | Variants | PSNR | SSIM
> -|-|-|-
> (Chitwan et al., 2021) | Channel 16, Res 1 | 21.12 | 0.693
> (Chitwan et al., 2021) | Channel 16, Res 2 | 18.15 | 0.652
> (Chitwan et al., 2021) | Channel 16, Res 3 | 19.86 | 0.678
> (Chitwan et al., 2021) | Channel 32, Res 1 | 19.49 | 0.671
> (Chitwan et al., 2021) | Channel 32, Res 2 | 19.00 | 0.667
> (Chitwan et al., 2021) | Channel 32, Res 3 | 19.33 | 0.660
> (Chitwan et al., 2021) | Channel 64, Res 1 | 17.99 | 0.670
> (Chitwan et al., 2021) | Channel 64, Res 2 | 21.10 | 0.690
> (Chitwan et al., 2021) | Channel 64, Res 3 | 20.10 | 0.687
> CDH (Ours) | --- | __24.24__ | __0.727__
>
>
> Compared to Palette (Chitwan et al., 2021), our method takes into account more halftoning task-specific properties, such as the blue-noise property of the halftone dithering process, and thus achieves better results. This validates the effectiveness of the proposed method.
>
> We greatly appreciate all your valuable suggestions for our paper, which make our work more comprehensive. Hope our
> response answered the question you were looking for. Thanks a lot!
>
> [1] Saharia Chitwan, Chan William, Chang Huiwen, et al. Palette: Image-to-image diffusion models. arXiv preprint, 2021.

---

### Official Review · Reviewer_kkXU · 2022-07-26

**Rating:** 6
**Confidence:** 3
**Soundness:** 3 good
**Presentation:** 2 fair
**Contribution:** 3 good

**Summary:**

The paper appears to beat the SOTA on inverse halftoning  with an approach that uses diffusion models. The details are not very transparent.

**Questions:**

See above

**Limitations:**

No comment

**Strengths And Weaknesses:**

The artifact reduction in inverse results is significant. In addition, the generated halftone images have a nice quality to them, so that might be useful, too. But the description of the details is difficult to read. Not enough is illustrates with images and the math is a it too abstract. When I read the title I assumed that the noising process in a diffusion model would reflect the dithering (going from the original image to a noise one by adjusting intensities until they are binary), but the method instead is using a traditional diffusion models to capture image distribution and a series of adjustments happens with meta-halftones (a new idea), which are not well illustrated. Nevertheless, the results look very good.

---

> ### Author Response · Authors · 2022-08-02
> **Author Responses to Reviewer 1 (kkXU)**
>
> Dear Reviewer:
> Thank you very much for your constructive comments on our paper! Our responses to your questions are below.
>
> In this paper, starting from the essence of the halftone process, we regard the black pixels in the halftone as particles in physical motion, and make them move under some certain guidance through the reverse diffusion process, so as to obtain the desired halftone distribution. In view of this, our proposed CDH method consists of a halftone dithering process (used to generate halftone images with multiple dithering patterns) and an inverse halftoning process (used to learn the mapping from halftone distributions to continuous-tone distributions). In order to avoid introducing redundant patterns and undesired artifacts during halftone generation, a meta-halftone guided network is proposed to incorporate the blue noise guidance into the halftone diffusion process.
>
> Thank you for your constructive comments, changing the image intensity through a diffusion process until halftone results are obtained is really an interesting study and we will discuss them in future work.
>
> Thanks again for your constructive comments!

---

### Meta-Review · Area_Chair_yvNP · 2022-09-01

**Recommendation:** Accept
**Confidence:** Certain

**Metareview:**

Most reviewers are positive about the paper. After rebuttal, a number of issues were addressed, and the additional evaluations provided in the authors' response should be published in the paper or supplemental. The clarity of the technical exposition and details can still be improved, but reviewers found the results better than SOTA and the ideas worth publishing

**Award:**

No

---

### Decision · Program_Chairs · 2022-09-14

Accept